# TSPAN15 interacts with BTRC to promote oesophageal squamous cell carcinoma metastasis via activating NF-κB signaling

Baozhu Zhang[1,2], Zhao Zhang[3], Lei Li[1], Yan-Ru Qin[4], Haibo Liu[5], Chen Jiang[6], Ting-Ting Zeng[1], Meng-Qing Li[1], Dan Xie[1], Yan Li[1], Xin-Yuan Guan [1,6] & Ying-Hui Zhu[1]

Beta-transducin repeat containing E3 ubiquitin protein ligase (BTRC) is crucial for the degradation of IκBα. Our previous transcriptome sequencing analysis revealed that *tetraspanin 15* (*TSPAN15*) was significantly upregulated in clinical oesophageal squamous cell carcinoma (OSCC) tissues. Here, we show that high TSPAN15 expression in OSCC tissues is significantly associated with lymph node and distant metastasis, advanced clinical stage, and poor prognosis. Elevated *TSPAN15* expression is, in part, caused by the reduction of miR-339-5p. Functional studies demonstrate that *TSPAN15* promotes metastatic capabilities of OSCC cells. We further show that TSPAN15 specifically interacts with BTRC to promote the ubiquitination and proteasomal degradation of p-IκBα, and thereby triggers NF-κB nuclear translocation and subsequent activation of transcription of several metastasis-related genes, including ICAM1, VCAM1, uPA, MMP9, TNFα, and CCL2. Collectively, our findings indicate that TSPAN15 may serve as a new biomarker and/or provide a novel therapeutic target to OSCC patients.

[1] Sun Yat-sen University Cancer Center, State Key Laboratory of Oncology in South China, Collaborative Innovation Center for Cancer Medicine, 510062 Guangzhou, China. [2] The People's Hospital of Baoan Shenzhen, The Affiliated Baoan Hospital of Southern Medical University, 518101 Shenzhen, China. [3] Department of Medical Oncology, Hainan Cancer Hospital, 570100 Haikou, China. [4] Department of Clinical Oncology, the First Affiliated Hospital, Zhengzhou University, 450052 Zhengzhou, China. [5] Key Laboratory for Major Obstetric Diseases of Guangdong Province, The Third Affiliated Hospital of Guangzhou Medical University, 510150 Guangzhou, China. [6] Department of Clinical Oncology, The University of Hong Kong, Hong Kong 999077, China. Correspondence and requests for materials should be addressed to Y.L. (email: liyan@sysucc.org.cn) or to X.-Y.G. (email: xyguan@hku.hk) or to Y.-H.Z. (email: zhuyh@sysucc.org.cn)

Oesophageal cancer is one of the most aggressive malignancies and rank as the sixth lethal cancer worldwide[1]. According to the latest estimate by the International Agency for Research on Cancer (IARC), approximately 455,800 new oesophageal cancer cases and 400,200 deaths occurred in 2012 worldwide[2]. The highest risk area extends from northern Iran through the Central Asian republics to northern China, where 90% of cases are histologically oesophageal squamous cell carcinoma (OSCC)[2]. Clinical methods for early diagnosis and treatment of OSCC are still limited, resulting in a 10% five-year survival rate[3]. Therefore, there is a pressing need to understand the molecular mechanisms underlying OSCC.

MicroRNAs (miRNAs), a class of small non-coding RNAs, mediate post-transcriptional gene regulation and are indispensable for physiological and pathological processes[4–6]. They bind to the 3′ untranslated region (3′UTR) of target messenger RNAs (mRNAs) to promote mRNA degradation and/or inhibit translation[7]. Over the past decade, miRNAs have emerged as critical regulators of the production of oncoproteins and tumor suppressor proteins in many cancers, including OSCC[8–10].

The transcription factor nuclear factor-κB (NF-κB) plays a crucial role in inflammation and innate immunity as well as in cancer initiation and porgression[11,12]. The prototypical inducible NF-κB complex, consisting of two distinct subunits, p50 and p65, is normally sequestered in the cytoplasm through interaction with the IκB family of inhibitory proteins[13]. Following stimulation, IκBα, the best characterized members of the IκB family, is phosphorylated by IκB kinases[14,15]. Such phosphorylation of IκBα triggers its degradation by ubiquitin-mediated proteolysis[16,17]. Ubiquitination of proteins involves the concerted action of an E1 ubiquitin-activating enzyme, E2 ubiquitin-conjugating enzymes, and E3 ubiquitin ligases that serves to bind both the E2 and the substrate. Beta-transducin repeat containing E3 ubiquitin protein ligase (BTRC, also known as β-TrCP and FWD1) is a member of the F-box and WD40 repeat family of proteins. Accumulating evidence has revealed that BTRC interacts with Skp1 and Cullin1 through its F-box domain to form an SCF complex for ubiquitination of phosphorylated IκBα (p-IκBα), thereby triggering translocation of NF-κB to the nucleus and activation of target genes[18–20].

To identify genetic alterations involved in the initiation and progression of OSCC, we exploited a transcriptome sequencing (RNA-Seq) to delineate differential gene expression in three pairs of OSCC and adjacent non-tumorous tissues. Overexpression of tetraspanin 15 (TSPAN15) was observed in all three OSCC tissues compared with their matched normal counterparts. We then employed a web serve GEPIA (Gene Expression Profiling Interactive Analysis) to validate this finding. GEPIA is an interactive web application for gene expression analysis based on the TCGA (the Cancer Genome Atlas) and the GTEx (Genotype-Tissue Expression) databases, using the output of a standard processing pipeline for RNA sequencing data[21]. The result demonstrates that TSPAN15 expression is obviously higher in esophageal cancer (ESCA) compared with normal tissues (Supplementary Fig. 1a), which is consistent with our RNA-Seq data. TSPAN15 gene, located at chromosome 10q22, belongs to the tetraspanin family. Tetraspanins are conserved small proteins characterized by four typical hydrophobic transmembrane domains (TM1–TM4), and one small (EC1) and one large extracellular loop (EC2)[22]. Generally, tetraspanins interact with other tetraspanins and neighboring membrane proteins to form tetraspanin-enriched microdomains (TEMs)[23]. Like lipid rafts, TEMs are considered as specific functional structures that promote protein−protein interactions, including membrane receptors, adhesion molecules, and signal transduction molecules[24]. To date, 33 tetraspanins family members have been identified in humans[25].

Downregulation of TSPAN27 (CD82) is strongly associated with poor outcomes of prostate cancer, colon cancer, cervical cancer, and ovarian cancer[26]. On the other hand, overexpression of TSPAN24 (CD151) was detected in breast, pancreatic, colorectal, and non-small-cell lung cancer (NSCLC) and predicted poor prognosis[26]. In light of these findings, the functions of tetraspanins appear to depend on the tissue context. TSPAN15 has been reported to influence the cellular trafficking and activity of ADAM metallopeptidase domain 10 (ADAM10), which cleaves many proteins including tumor necrosis factor-alpha and E-cadherin[27]. However, the expression and the function of TSPAN15 in OSCCs are still unclear.

In the present study, we find that TSPAN15 is frequently upregulated in OSCC. Functional studies demonstrate that TSPAN15 enhances both in vitro and in vivo metastatic capabilities of OSCC cells. Mechanistic investigations suggest that TSPAN15 interacts with BTRC and enhances the latter's ubiquitin activity for p-IκBα, thus facilitating the degradation of p-IκBα and subsequent activation of NF-κB nuclear translocation, thereby promoting OSCC metastasis.

## Results

**Overexpression of TSPAN15 is frequently detected in OSCC.** Our previous RNA-seq analysis revealed that TSPAN15 was overexpressed in three tested OSCC tumor tissues compared to their corresponding non-tumor tissues. Upregulated mRNA expression level of TSPAN15 (defined as >2-fold increase) was initially detected in 28/46 (60.9%) of OSCC tissues compared with the corresponding non-tumorous tissues ($P < 0.001$, paired Student's $t$-test; Fig. 1a). In addition to clinical OSCC specimens, elevated TSPAN15 expression was also observed in 8/10 OSCC cell lines compared with an immortalized oesophageal epithelial cell line (NE1) (Fig. 1b).

**Clinical significance of TSPAN15 expression in OSCC.** To explore the clinical significance of TSPAN15 expression in OSCC, a tissue microarray containing 300 pairs of primary OSCCs and their matched non-tumor tissues was studied by immunohistochemistry (IHC). Informative results of IHC staining were obtained from 266 pairs of OSCCs. TSPAN15 was seldom detected in the normal oesophageal epithelium, but with increased levels in the OSCC sectors (Fig. 1c). Overexpression of TSPAN15 was detected in 169/266 (63.5%) of informative OSCC tumor tissues compared with the corresponding non-tumor tissues. According to ROC curve analysis, the optimum cutoff value for TSPAN15 staining index was 5. Therefore, TSPAN15 expression in OSCC was divided into two groups: high group ($\geq 5$, $n = 115$) and low group ($< 5$, $n = 151$). As shown in Table 1, high expression of TSPAN15 was significantly associated with lymph node metastasis ($P < 0.001$), distant metastasis ($P = 0.003$), and advanced clinical stage ($P = 0.001$). Kaplan−Meier survival curve analysis demonstrated that patients with high TSPAN15 expression have lower survival rate compared with those with low TSPAN15 expression (log-rank test, $P < 0.001$, Fig. 1d). In the univariate analysis, tumor location ($P < 0.001$), differentiation ($P = 0.001$), tumor invasion ($P < 0.001$), lymph node metastasis ($P < 0.001$), distant metastasis ($P = 0.002$), clinical stage ($P < 0.001$), and TSPAN15 expression ($P < 0.001$) were the statistically significant predictors for a patient's overall survival (Table 2). Multivariate Cox proportional regression analysis further revealed that tumor location ($P < 0.001$), differentiation ($P = 0.001$), and TSPAN15 expression ($P = 0.007$) are independent prognostic factors for the overall survival of OSCC patients (Table 2).

**TSPAN15 is negatively regulated by miR-339**. By using an online bioinformatics database (TargetScan, http://www.targetscan.org), miR-339-5p was identified as a putative miRNA targeting *TSPAN15*. Downregulation of miR-339-5p (defined as >2-fold decrease) was detected in 30/46 (65.2%) of OSCC tumors compared with corresponding nontumor tissues ($P < 0.001$, paired Student's *t*-test; Fig. 1e). Further, linear regression analysis showed that the expression of miR-339-5p was negatively correlated with *TSPAN15* expression in 46 OSCC cases ($R = -0.493$, $P < 0.001$; Fig. 1f). MiR-339-5p mimic or control miRNA (miR-Ctr) was then transfected into KYSE30 and KYSE140 to evaluate the influence of miR-339-5p on the *TSPAN15* mRNA expression level. The mRNA level of *TSPAN15* was significantly reduced after the introduction of miR-339-5p (Fig. 1g), indicating that miR-339-5p promotes the degradation of *TSPAN15* mRNA. To verify whether *TSPAN15* is a direct target of

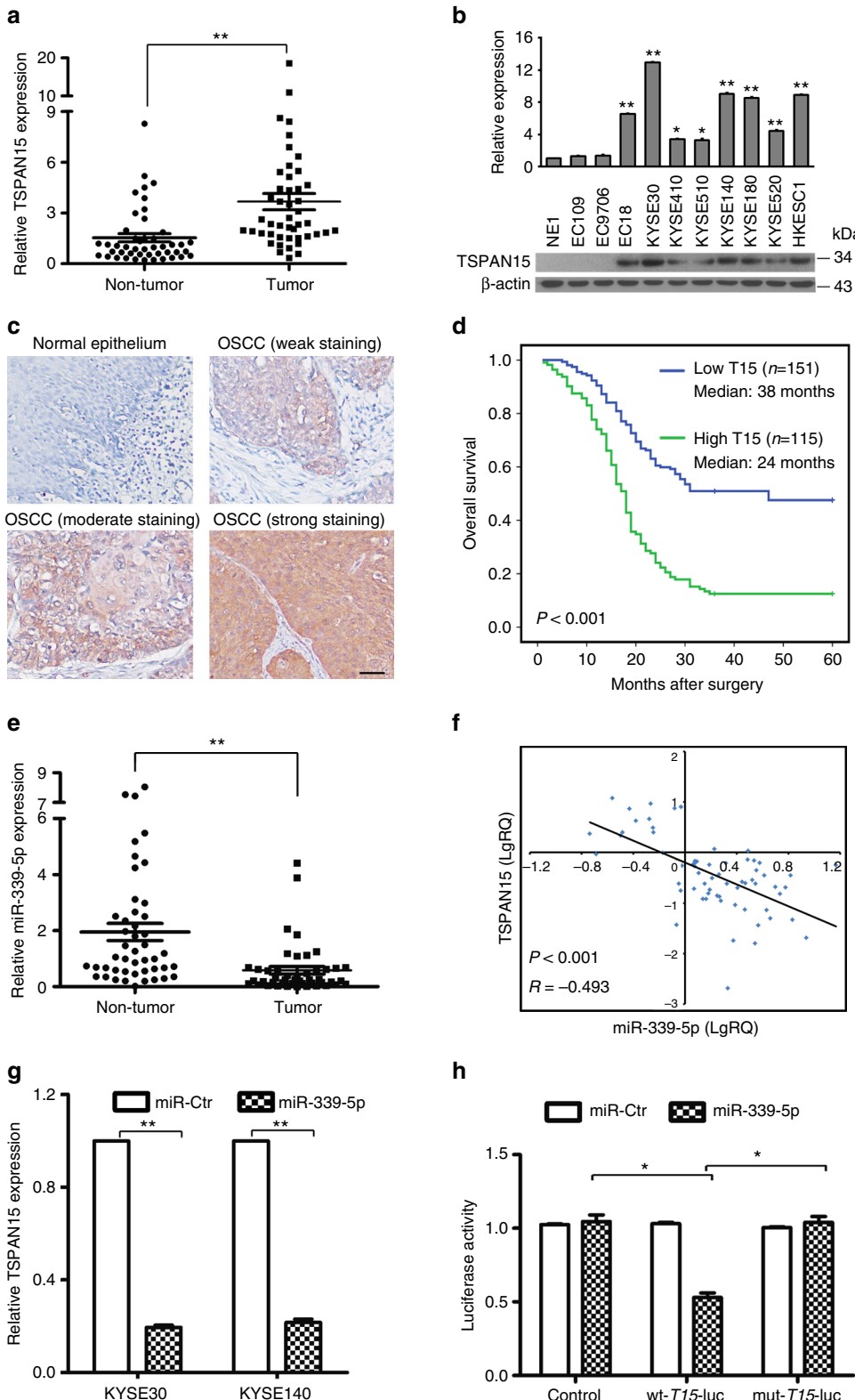

**Fig. 1** TSPAN15 is overexpressed in OSCC and negatively regulated by miR-339. **a** Relative expression levels of *TSPAN15* detected by qRT-PCR in 46 OSCC tissues compared with their adjacent non-tumor tissues. Expression of *β-actin* was used as an endogenous control. **P < 0.001, paired Student's *t*-test. **b** qRT-PCR and western blot analysis of TSPAN15 expression in one immortalized oesophagus cell line (NE1) and ten OSCC cell lines. Expression of β-actin was used as an endogenous control. The data are represented as the mean ± s.d. of three independent experiments. *P < 0.01, **P < 0.001, Student's *t*-test. **c** Representative of TSPAN15 expression in normal epithelium and OSCC tumor tissues detected by immunostaining with anti-TSPAN15 antibody (brown). The slide was counterstained with hematoxylin. Scale bar, 50 μm. **d** Kaplan−Meier analysis revealed that TSPAN15 overexpression was related with poorer overall survival of OSCC patients. *P* < 0.001, log-rank test. **e** Higher expression levels of miR-339 detected by qRT-PCR in 46 OSCC tissues compared with their adjacent non-tumor tissues. **P < 0.001, paired Student's *t*-test. **f** Analysis showing linear regressions and significant Pearson correlations of miR-339-5p with *TSPAN15* in 46 pairs of OSCC samples. **g** The mRNA levels of *TSPAN15* decreased when miR-339-5p mimics was transfected into KYSE30 and KYSE140 cells compared with control cells. The data are represented as the mean ± s.d. of three independent experiments. **P < 0.001, Student's *t*-test. **h** Luciferase assay confirmed that miR-339-5p targeted the 3′-UTR region of *TSPAN15*. MiR-339-5p-expressing cells and control cells were co-transfected with wt-T15-luc, mu-T15-luc or empty vector with pRL-TK, and relative luciferase activity was detected. The data are represented as the mean ± s.d. of three independent experiments. *P < 0.01, Student's *t*-test

## Table 1 Clinicopathological correlation of TSPAN15 expression in OSCC

| Clinical features | Cases | TSPAN15 expression | | P value |
| --- | --- | --- | --- | --- |
| | | Low group (%) | High group (%) | |
| Age(years old) | | | | 0.388 |
| ≤59 | 131 | 78 (59.5%) | 53 (40.5%) | |
| >59 | 135 | 73 (54.1%) | 62 (45.9%) | |
| Gender | | | | 0.323 |
| Male | 148 | 80 (54.1%) | 68 (45.9%) | |
| Female | 118 | 71 (60.2%) | 47 (39.8%) | |
| Location | | | | 0.245 |
| Upper | 63 | 30 (47.6%) | 33 (52.4%) | |
| Middle | 183 | 109 (59.6%) | 74 (40.4%) | |
| Lower | 20 | 12 (60.0%) | 8 (40.0%) | |
| Differentiation | | | | 0.421 |
| Grade 1 | 63 | 39 (61.9%) | 24 (38.1%) | |
| Grade 2 | 134 | 77 (57.5%) | 57 (42.5%) | |
| Grade 3 | 69 | 35 (69.6%) | 34 (30.4%) | |
| LN metastasis | | | | **<0.001** |
| N0 | 149 | 102 (68.5%) | 47 (31.5%) | |
| N1 | 117 | 49 (41.9%) | 68 (58.1%) | |
| Distant metastasis | | | | **0.003** |
| M0 | 256 | 150 (58.6%) | 106 (41.4%) | |
| M1 | 10 | 1 (10.0%) | 9 (90.0%) | |
| Clinical stage | | | | **0.001** |
| Early (I−II) | 177 | 111 (62.7%) | 66 (37.3%) | |
| Advanced (III−IV) | 89 | 40 (44.9%) | 49 (55.1%) | |

Statistical significance (*P* < 0.05) is shown in bold

miR-339-5p, we constructed luciferase reporter plasmids carrying wild-type *TSPAN15* 3′-UTR or mutant *TSPAN15* 3′-UTR. Empty luciferase reporter plasmid was used as a control. The relative luciferase activity in EC109 cells co-transfected with miR-339-5p mimic and wt-T15-luc was sharply inhibited, while the luciferase activity was unaffected in those cells co-transfected with miR-339-5p mimic and mut-T15-luc (Fig. 1h). Collectively, these data support that miR-339-5p reduces *TSPAN15* expression by directly targeting the 3′-UTR of *TSPAN15* mRNA.

**Ectopic overexpression of *TSPAN15* enhances tumor metastasis**. To explore its role in tumorigenicity, *TSPAN15* was stably introduced into EC109 and EC9706 (109-T15 and 9706-T15) cells. Empty vector-transfected cells (109-Vec and 9706-Vec) were used as controls. The efficiency of *TSPAN15* transfection was assessed by qRT-PCR and western blot analysis (Fig. 2a). The tumorigenic ability of *TSPAN15* was assessed by XTT cell growth

assay, foci formation assay, soft agar assay and xenograft tumor mouse model. However, no significant differences were observed between *TSPAN15*-expressing and control cells (Supplementary Fig. 1b–e). Wound-healing and transwell invasion assays were further carried out to investigate the effects of *TSPAN15* over-expression on metastasis. Compared with control cells, *TSPAN15*-expressing cells (109-T15 and 9706-T15) demonstrated elevated migratory and invasive capabilities (Fig. 2b and Supplemental Fig. 2a). To confirm the in vivo role of *TSPAN15* overexpression on metastasis, *TSPAN15*-expressing or empty vector-transfected cells were intravenously injected into nude mice to mimic cell metastasis through circulation. At 8 weeks after injection, mice were killed, and their lungs were harvested. More pulmonary metastatic nodules were observed in mice injected with 109-T15 or 9706-T15 cells compared with the mice injected with control cells (14.33 ± 2.50 vs. 4.67 ± 1.75, *P* < 0.01 for EC109; 11.50 ± 2.1 vs. 2.33 ± 1.03, *P* < 0.01 for EC9706, respectively, Student's *t*-test, Fig. 2c). Histological studies confirmed that the pulmonary lesions were caused by extravasation and subsequent tumor growth of OSCC cells (Fig. 2c).

**Knockdown of *TSPAN15* inhibits tumor metastasis**. To determine whether endogenous *TSPAN15* affects cell migration and invasion, two specific short hairpin RNAs (shRNA) targeted *TSPAN15* (sh*T15*-1 and sh*T15*-2) were transfected into KYSE30 cells (Fig. 2d). A scrambled shRNA (shNC) was used as a negative control. Knockdown of *TSPAN15* in KYSE30 cells led to a significant reduction in cell mobility and invasiveness both in vitro and in vivo (Fig. 2e, f; Supplementary Fig. 2b).

**TSPAN15 interacts with BTRC**. To identify candidate proteins that interact and collaborate with TSPAN15, protein pulldown assay was performed. One of the most abundant and specific band was separated and identified as beta-transducin repeat containing E3 ubiquitin protein ligase (BTRC) by liquid chromatography-mass spectrometry (LC/MS-MS) analysis (Fig.3a, indicated by red box). Immunoprecipitation (IP) assays confirmed that BTRC existed in complexes precipitated with antibody against TSPAN15 compared with control IgG (Fig. 3b, upper). Endogenous TSPAN15's binding with BTRC was further validated by IP assay with antibody against BTRC (Fig. 3b, lower). To further confirm whether TSPAN15 colocalizes with BTRC, a flag-tagged *TSPAN15* plasmid was transfected into EC109 and EC9706 cells. IF staining showed that flag-tagged *TSPAN15* extensively colocalized with BTRC in the cytoplasm (Fig. 3c). Additionally, western blotting demonstrated that the binding of TSPAN15 to BTRC did not affect their respective protein levels (Supplementary Fig. 2c).

**Table 2 Cox proportional hazard regression analyses for overall survival**

| Clinical features | Univariate analysis | | Multivariate analysis | |
|---|---|---|---|---|
| | HR[a] (95% CI[b]) | P value | HR (95% CI) | P value |
| Gender | 0.881 (0.654–1.187) | 0.406 | | |
| Age | 1.294 (0.964–1.736) | 0.086 | | |
| Grade | 0.994 (0.667–1.483) | 0.978 | | |
| Location | 0.572 (0.434–0.755) | <0.001 | 0.484 (0.353–0.664) | <0.001 |
| Differentiation | 1.411 (1.142–1.742) | 0.001 | 1.461 (1.169–1.827) | 0.001 |
| Tumor invasion | 1.618 (1.245–2.103) | <0.001 | 1.283 (0.920–1.790) | 0.142 |
| LN metastasis | 2.080 (1.548–2.795) | <0.001 | 1.520 (0.922–2.505) | 0.101 |
| Distant metastasis | 2.738 (1.437–5.217) | 0.002 | 0.618 (0.251–1.525) | 0.297 |
| Clinical stage | 2.319 (1.718–3.130) | <0.001 | 1.534 (0.934–2.518) | 0.091 |
| T15 expression | 1.868 (1.383–2.524) | <0.001 | 1.538 (1.127–2.098) | 0.007 |

[a] Hazard ratio
[b] Confidence interval

**TSPAN15 promotes the degradation of p-IκBα.** Considering that BTRC has been reported to serve as a component of SCF complex that participates in the ubiquitin-mediated proteolysis of p-IκBα, we initially validate the effects of BTRC on the degradation of p-IκBα. As expected, knockdown of *BTRC* in KYSE30 cells prohibited the degradation of p-IκBα (Supplementary Fig. 2d). Subsequently, we investigated whether the interaction between TSPAN15 and BTRC influenced the degradation of p-IκBα. Intriguingly, decreased expression of p-IκBα was observed in 109-*T15* and 9706-*T15* cells compared with their control cells, respectively (Fig. 3d). By contrast, the protein level of p-IκBα increased in *TSPAN15*-knockdown cells (Fig. 3d). To assess whether TSPAN15 promotes the BTRC-mediated degradation of p-IκBα, *TSPAN15*-expressing cells and control cells were further treated with a pro-teasome inhibitor MG132 (Selleck Chemicals, Houston, TX). Six hours after introduction of MG132 (10 μmol L$^{-1}$), cell lysates were obtained and subjected to immunoprecipitation with an anti-p-IκBα antibody followed by ubiquitin immunoblotting. Increased level of poly-ubiquitinated p-IκBα was observed in *TSPAN15*-expressing cells compared with control cells (Fig. 3e). In contrast, knockdown of *TSPAN15* reduced the accumulation of poly-ubiquitinated p-IκBα in KYSE30 compared with the control cells (Fig. 3e). In addition, treatment of *TSPAN15*-expressing cells with a protein synthesis inhibitor cycloheximide (CHX) resulted in a notably shorter half-life for p-IκBα than in control cells (Fig. 3f, upper). Conversely, knockdown of *TSPAN15* in KYSE30 cells yielded opposing effects (Fig. 3f, lower). These results imply that TSPAN15 promotes the degradation of p-IκBα by ubiquitin-mediated proteolysis.

**TSPAN15 activates NF-κB signaling.** Degradation of IκBα can lead to nuclear translocation of various NF-κB complexes, pre-dominantly the p50/p65 dimer. As expected, IF staining indicated that NF-κB p65 was located predominantly in the nucleus of *TSPAN15*-overexpressing cells, whereas mainly in the cytoplasm of control cells. Conversely, knockdown of *TSPAN15* impaired p65 nuclear translocation (Fig. 4a). Subsequent qRT-PCR and western blot analysis showed that several NF-κB-regulated metastasis-related genes, including intercellular adhesion mole-cule 1 (ICAM1), vascular cell adhesion molecule 1 (VCAM1), urokinase-type plasminogen activator (uPA), matrix metallo-peptidase 9 (MMP9), tumor necrosis factor (TNFα), and C-C motif chemokine ligand 2 (CCL2) were markedly increased in 109-*T15* and 9706-*T15* cells compared with their control cells (Fig. 4b, c). Conversely, knockdown of *TSPAN15* in KYSE30 cells yielded opposing effects (Fig. 4b, c). Additionally, IHC was used to measure the expression of these proteins in mice pulmonary metastatic nodules. Strong staining of the above-mentioned proteins was observed in 109-*T15*-derived lung nodes compared with 109-*Vec*-derived lung nodes (Fig. 4d). To further validate the importance of TSPAN15 in activating NF-κB signaling in clinical samples, the expression of TSPAN15 and p65 was initially assessed in 67 OSCC cases. IHC staining revealed that nuclear localization of p65 was consistently detected in the OSCC with high TSPAN15 expression, whereas cytoplasmic p65 was observed in the cases with low TSPAN15 expression (Fig. 4e). p65 expression was further examined by IHC using the same TMA. Informative results were obtained from 248 cases including the cases detected by TSPAN15 mentioned above. Correlation study showed that high TSPAN15 expression was significantly asso-ciated with nuclear localization of p65 ($P < 0.001$; Table 3).

**TSPAN15-induced NF-κB activation is BTRC dependent.** To verify whether TSPAN15 activates the NF-κB signaling in a BTRC-dependent manner, RNAi was used to knockdown *BTRC* expression in 109-*T15* cells (Fig. 5a). Wound-healing and invasion assays showed that *BTRC* knockdown significantly impaired cell migration and invasive capabilities in 109-*T15* cells (Fig. 5b and Supplementary Fig. 2e). Moreover, deletion of *BTRC* in 109-*T15* cells efficiently suppressed p65 nuclear translocation (Fig. 5c), as well as the expression of ICAM1, VCAM1, uPA, MMP9, TNFα, and CCL2 (Fig. 5d). Taken together, these results indicate that TSPAN15 activates the NF-κB signaling via inter-acting with BTRC.

**NF-κB inhibitor abrogates TSPAN15-induced OSCC metas-tasis.** To evaluate the effects of NF-κB inhibitor on *TSPAN15*-expressing cells, JSH-23 was applied to inhibit lipopolysacchar-ides and cytokine-induced nuclear translocation of p65. IF staining showed that JSH-23 strongly prevented p65 from entering the nucleus in 109-*T15* and 9706-*T15* cells, whereas few changes were observed in cells treated with DMSO (Fig. 5e, upper). In addition, caffeic acid phenethyl ester (CAPE) was applied to further confirm the effects of NF-κB inhibitor. IF staining revealed that CAPE inhibited p65 to enter the nucleus of 109-*T15* and 9706-*T15* cells compared to the cells with DMSO treatment (Fig. 5e, lower). In vitro and in vivo assays showed that both JSH-23 and CAPE efficiently abrogated *TSPAN15*-induced elevated cell mobility and invasiveness (Fig. 5f, g).

## Discussion

TSPAN15 belongs to the TspanC8 subfamily of tetraspanins, which are characterized by the presence of eight cysteines in the large extracellular domain[27]. The other family members are

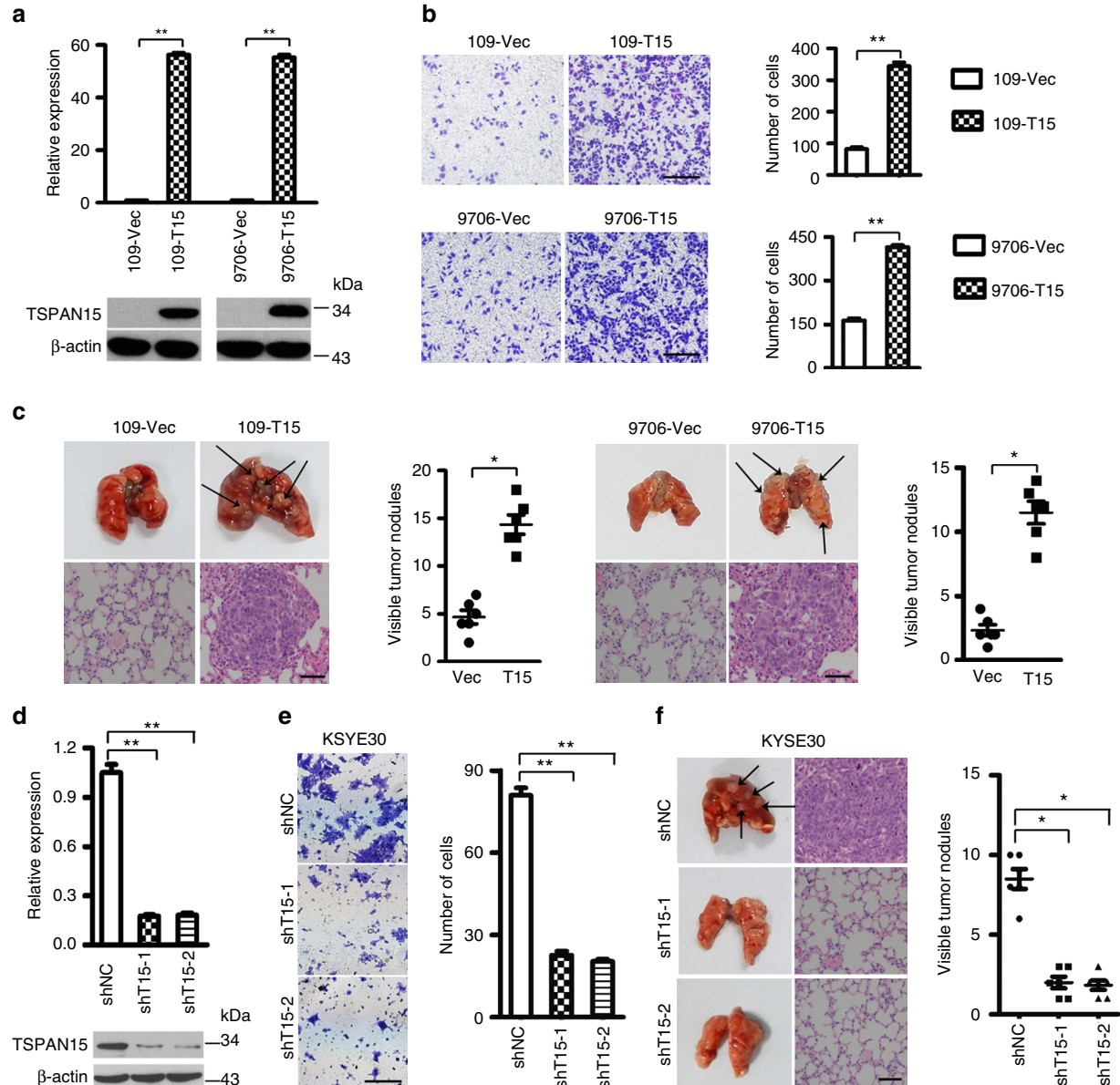

**Fig. 2** TSPAN15 promotes OSCC metastasis. **a** The mRNA and protein levels of TSPAN15 were detected by qRT-PCR and western blot analysis in *TSPAN15*- or vector-transfected cells. β-actin was used as an endogenous control. The data are represented as the mean ± s.d. of three independent experiments. **$P < 0.001$, Student's *t*-test. **b** Representatives and summary of invasion assay showing that overexpression of *TSPAN15* promoted cell invasion. Scale bar, 200 μm. The values are expressed as the mean ± s.d. of three independent experiments. **$P < 0.001$, Student's *t*-test. **c** Representatives and summary of pulmonary metastatic nodules formed in nude mice intravenously injected with *TSPAN15*- or Vec-transfected cells. Black arrow indicates the pulmonary metastatic nodules. Pulmonary nodules invaded by tumor cells were confirmed by H&E staining (scale bar, 50 μm). The values are expressed as the mean ± s.d. of six mice. *$P < 0.01$, Student's *t*-test. **d** Two shRNAs (sh*T15*-1 and sh*T15*-2) against *TSPAN15* effectively silenced *TSPAN15* expression as detected by qRT-PCR and western blotting. Negative control shRNA (shNC) and β-actin were used as negative and endogenous controls, respectively. The data are represented as the mean ± s.d. of three independent experiments. **$P < 0.001$, ANOVA with post hoc test. **e** Representatives and summary of invasion assay showing that knockdown of *TSPAN15* effectively inhibited cell invasion. Scale bar, 200 μm. The values are expressed as the mean ± s.d. of three independent experiments. **$P < 0.001$, ANOVA with post hoc test. **f** Representatives and summary of pulmonary metastatic nodules formed in nude mice intravenously injected with sh*T15*-1, sh*T15*-2 or shNC-transfected KYSE30 cells. Black arrow indicates the pulmonary metastatic nodule. Corresponding H&E staining images were also displayed. Scale bar, 50 μm. The values are expressed as the mean ± s.d. of six mice. *$P < 0.01$, ANOVA with post hoc test

TSPAN5, TSPAN10, TSPAN14, TSPAN17, and TSPAN33[27]. Compared with other tetraspanins, TspanC8 family members exhibit evolutionarily conservation from invertebrates to mammals[28]. Overexpression of *TSPAN10* was found in metastatic uterine leiomyosarcoma[29]. Moreover, single nucleotide variations in exonic region of *TSPAN10* were detected in metastatic melanoma. Genetic alteration of TSPAN14 in grade 1, stage I of NSCLC indicated that structural change of TSPAN14 might

promote NSCLC[30]. Given TSPAN15 shares the same structural features with the above-mentioned TspanC8 tetraspanins, it may play a role in cancer initiation and development. In this study, we demonstrate that TSPAN15 is upregulated in OSCC samples compared with their matched non-tumor counterparts. Subsequent clinical analysis reveals that overexpression of TSPAN15 is associated with lymph node and distant metastasis, advanced clinical stage, and poor prognosis of OSCC patients.

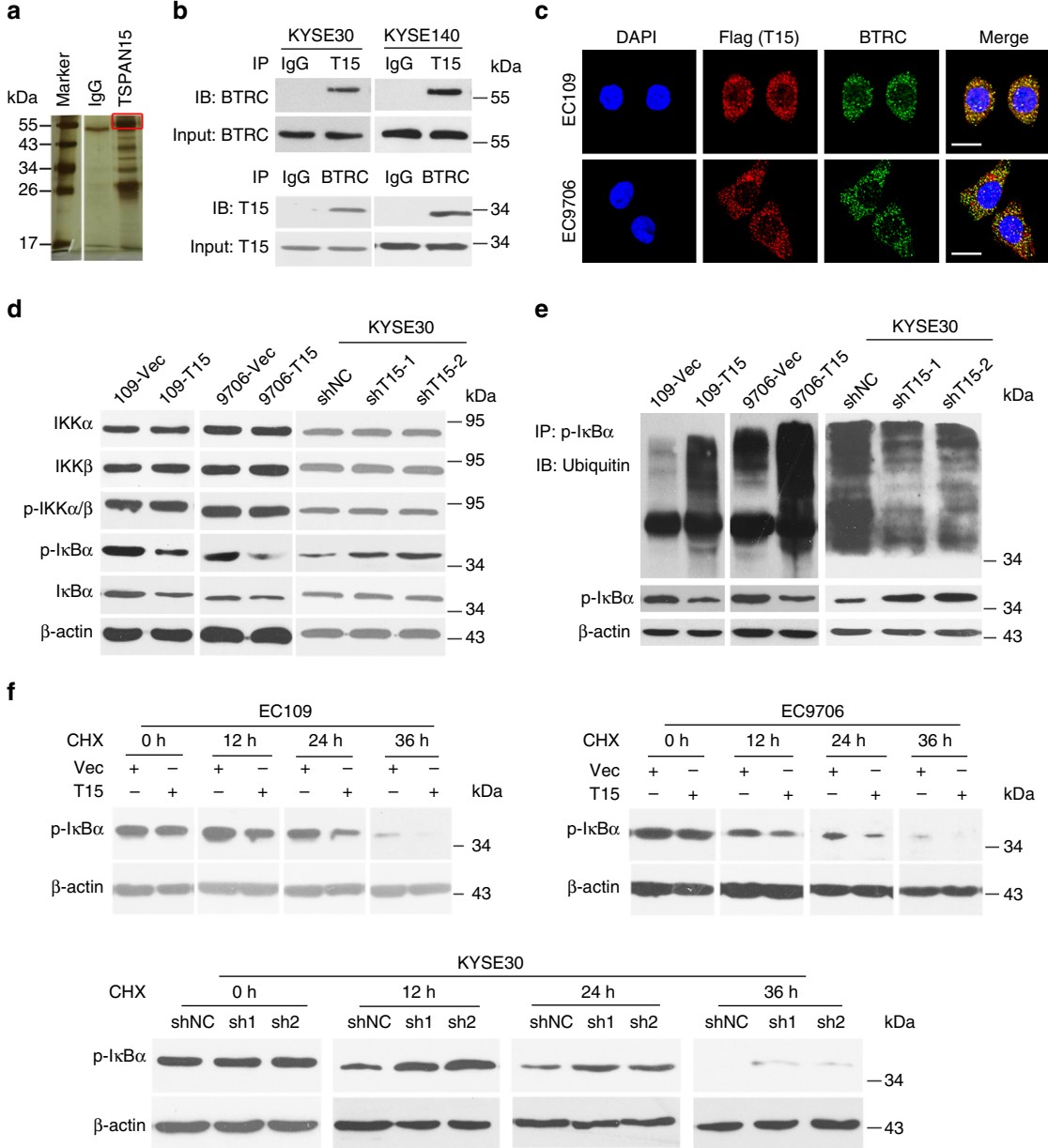

**Fig. 3** TSPAN15 interacts with BTRC to boost p-IκBα degradation. **a** Protein pulldown experiment for the identification of proteins associated with TSPAN15 by incubation of TSPAN15-antibody with protein extracts from KYSE30 cells. The red box indicates the most abundant additional band that was present when the protein extracts were incubated with TSPAN15 as compared to IgG. **b** Immunoprecipitation assay showed that TSPAN15-antibody could pull down BTRC (upper). Correspondingly, BTRC-antibody also pulled down TSPAN15 (lower). **c** IF showed that colocalization of flag-TSPAN15 (red) and BTRC (green). Nuclei were stained with DAPI (blue). Scale bar, 20 μm. **d** Overexpression of *TSPAN15* decreased the protein level of p-IκBα (left). Conversely, p-IκBα increased in *TSPAN15*-knockdown cells (right). **e** MG-132-induced accumulation of polyubiquitinated p-IκBα was influenced by overexpression or knockdown of *TSPAN15*. β-actin was used as a loading control. **f** CHX assays showed that overexpression or knockdown of *TSPAN15* affected the degradation rate of p-IκBα. β-actin was used as a loading control

Although miRNAs can serve either as oncogenes or tumor-suppressor genes, several studies have illustrated that miRNA expression is globally attenuated in tumor cells compared with normal tissue, implying that miRNA biogenesis might be suppressed during carcinogenesis. Recent studies have shown that miR-339-5p is downregulated and serves as a tumor suppressor gene in various cancers, including ovarian cancer[31], colorectal cancer[32], small-cell lung cancer[33], non-small cell lung cancer[34], breast cancer[35], and hepatocellular carcinoma[36]. Indeed, we found that miR-339-5p was downregulated in OSCC clinical samples, which was significantly associated with increased

*TSPAN15* expression. Furthermore, we prove that miR-339-5p directly targets 3′-UTR of *TSPAN15* to suppress *TSPAN15* expression at the transcriptional level, suggesting that the absence of miR-339-5p is one of mechanisms responsible for upregulation of *TSPAN15*. Nevertheless, further investigation is still needed to unravel the precise mechanism underlying the regulation of *TSPAN15* in OSCC.

Functional studies in vitro and in vivo revealed that *TSPAN15* promoted cell migration, cell invasion, and tumor metastasis. Cancer metastasis is a complicated process that requires cancer cells to adhere to and then penetrate the vascular endothelium at

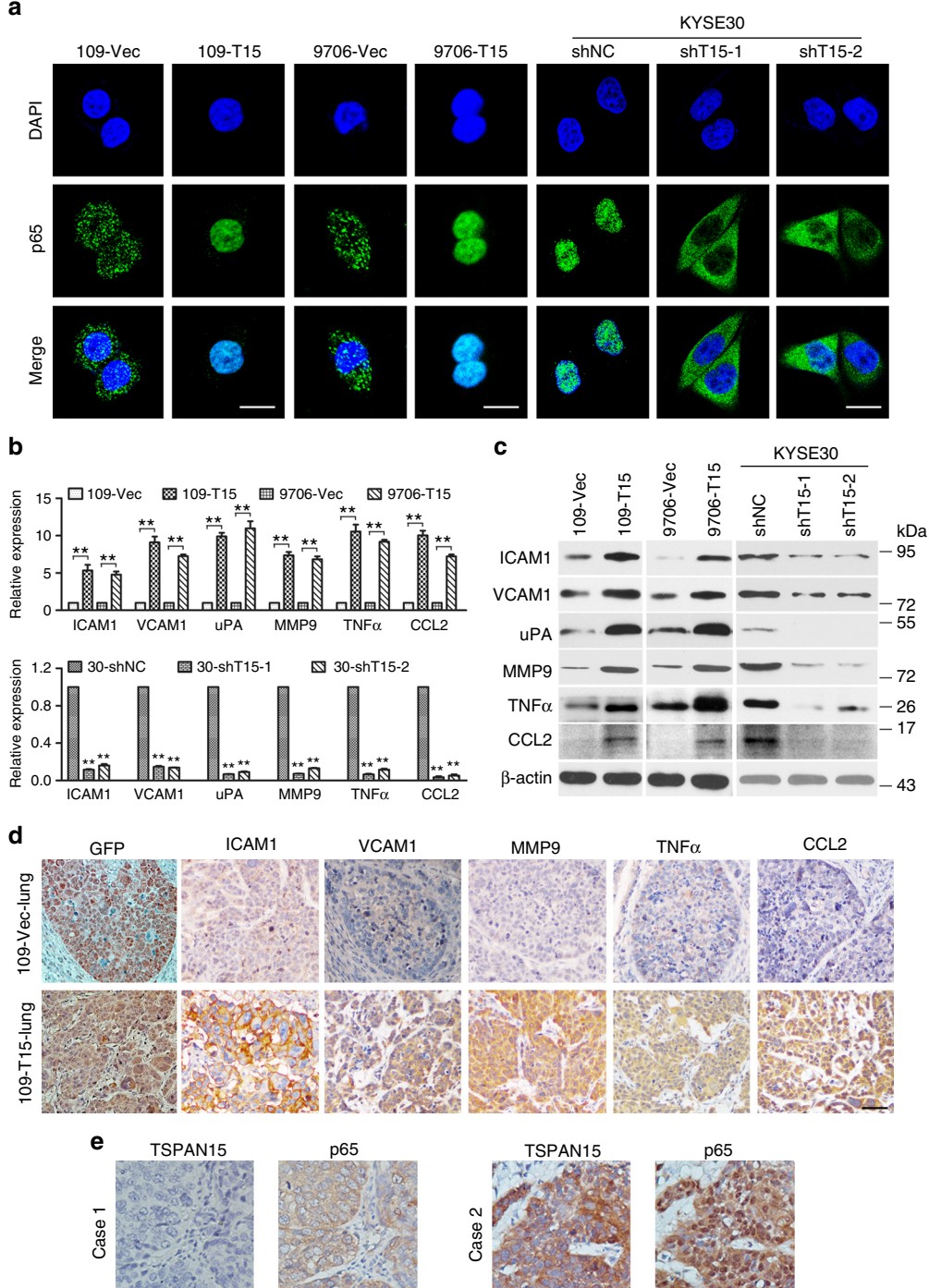

**Fig. 4** Overexpression of *TSPAN15* activates NF-κB signaling and enhances the expression of metastasis-related genes. **a** Overexpression of *TSPAN15* increased nuclear location of NF-κB p65 (green), while knockdown of *TSPAN15* reduced nuclear location of NF-κB p65 (green). Nuclei were counterstained with DAPI. Scale bar, 20 μm. **b** qRT-PCR showed elevated transcript levels of ICMA1, VCAM1, uPA, and MMP9 in *TSPAN15*-expressing cells (upper). In contrast, knockdown of *TSPAN15* yielded opposing effects (lower). *β-actin* was set as an internal control. The data are represented as the mean ± s.d. of three independent experiments. **P < 0.001, Student's *t*-test; ANOVA with post hoc test. **c** Protein levels of ICMA1, VCAM1, uPA MMP9, TNFα, and CCL2 were compared by western blot analysis. β-actin served as a loading control. **d** Representative images of IHC staining of GFP, ICAM1, VCAM1, MMP9, TNFα, and CCL2 in lung sections derived from mice injected with *TSPAN15*-expressing cells or control cells. Scale bar, 50 μm. **e** Representative pictures of TSPAN15 and p65 in serial section of clinical samples. Scale bar, 50 μm

the metastatic sites[37]. Cell adhesion molecules (CAMs) and matrix metalloproteinases (MMPs) are important proteins that contribute to the escape of cancer cells from blood circulation[38–40]. In addition, both inflammatory chemokine CCL2 and inflammatory factor TNFα have been identified as

crucial promoters of cancer progression and metastasis[41,42]. Interestingly, the genes encoding VCAM1, ICAM1, MMP9, CCL2, and TNFα are the downstream targets of the NF-κB transcription factor. The systematic control of the NF-κB pathway relies on the unique property of its major inhibitor, IκBα[43].

**Table 3 Association of T15 expression and nuclear localization of p65 in OSCC**

| TSPAN15 expression | Cases | Nuclear localization of p65 | | P value |
|---|---|---|---|---|
| | | Low (%) | High (%) | |
| | | | | <0.001 |
| Low | 145 | 115 (79.3%) | 30 (20.7%) | |
| High | 103 | 29 (28.2%) | 74 (71.8%) | |

In the canonical pathway of NF-κB activation, ubiquitin-mediated proteasomal degradation of IκBα releases p65/p50 protein dimers to translocate to the nucleus, and subsequently induce transcription of target genes[44,45]. BTRC is an E3 ubiquitin ligase, which binds specifically to phosphorylated IκBα and mediates its ubiquitination, thereby leading to its degradation by the 26S proteasome[18]. In the present study, we show that TSPAN15 binds to BTRC and enhances its enzyme activity to ubiquitinylate p-IκBα, thus promoting NF-κB heterodimers to translocate to the nucleus and activate the transcription of metastasis-related genes.

Activation of NF-κB signaling pathway has been observed in a variety of human cancers[46]. Intriguingly, we found that two chemical inhibitors of NF-κB, JSH-23 and CAPE, could efficiently eliminated abrogated TSPAN15-induced elevated cell mobility and invasiveness, suggesting that inhibition of NF-κB may be an effective treatment strategy to the OSCC patients with high TSPAN15 expression. However, further investigation is required to confirm this finding.

Besides, as a specific binding partner of ADAM10[47], TSPAN15 promotes ADAM10-mediated shedding of a range of cell surface proteins, including Notch[48], Eph[49] and certain epidermal growth factor receptor ligands[50]. Accumulating evidence has shown that ADAM10 is frequently overexpressed in a variety of cancers, playing an important role in cancer pathogenesis, progression, metastasis and invasion[51–53]. In line of these findings, TSPAN15 may exert its tumorigenic roles in an ADAM10-dependent manner, in addition to the activation NF-κB signaling. In summary, we revealed that TSPAN15 is a novel oncogene, which plays an important role in the pathogenesis and metastasis of OSCC and may serve as a valuable prognostic marker for OSCC patients. Better understanding of the oncogenic mechanisms of TSPAN15 may provide a new therapeutic strategy for OSCC.

## Methods

**Clinical samples and cell lines**. Forty-six pairs of fresh primary OSCCs and the corresponding non-tumorous tissues were collected immediately after surgical resection at Linzhou Cancer Hospital (Henan, China). A number of 300 formalin-fixed and paraffin-embedded OSCCs and their adjacent non-tumorous tissue samples were also obtained from Linzhou Cancer Hospital. All patients signed informed consent. No patients recruited in the study received preoperative treatments. All OSCC samples used in this study were authorized by the Committees for Ethical Review of Research at Sun Yat-sen University Cancer Center (Guangzhou, China) and Zhengzhou University (Zhengzhou, China). Six OSCC cell lines (KYSE30, KYSE140, KYSE180, KYSE410, KYSE510, and KYSE520) were obtained from DSMZ (Braunschweig, Germany), the German Resource Centre for Biological Material[54]. Four Chinese OSCC cell lines (EC18, EC109, EC9706, and HKESC1) were kindly provided by Professors Tsao and Professor Srivastava (The University of Hong Kong)[55]. All ten human OSCC cells were cultured in DMEM supplemented with 10% FBS. All cell lines used in this study were regularly authenticated by morphologic observation and tested for absence of mycoplasma contamination (MycoAlert, Lonza).

**Tissue microarray (TMA) and immunohistochemistry (IHC)**. Three hundred cases of OSCC were selected for the TMA construction. TMA tissue specimens for each case were composed of one sample from normal oesophageal epithelium and one sample from primary OSCC. Briefly, a TMA instrument (Beecher Instruments, Silver Spring, Maryland) was employed to acquire a cylindrical tissue core with a

diameter of 0.6 mm from the donor block and transfer these biopsy samples to the recipient block at defined array positions. Using a microtome, multiple sections (5 μm thick) were cut from the TMA blocks to generate TMA slides for immuno-histochemical analyses. IHC was performed according to a standard streptavidin-biotin-peroxidase complex method. Paraffin-embedded, formalin-fixed sections were dewaxed and blocked in hydrogen peroxide/methanol solution. Antigen retrieval was performed by pressure-cooking the sample in 0.08% citrate buffer for 30 min. Slides were incubated with primary antibodies at 4 °C overnight. Staining results were visualized by sequential incubations of slides with an Envision detection system (DAKO), and the nucleus was counterstained using Meyer's hematoxylin. Immunohistochemistry staining was assessed by two independent pathologists with no prior knowledge of patient characteristics. Discrepancies were resolved by consensus. The staining extent score was on a scale of 0–4, corresponding to the percentage of immunoreactive tumor cells (0%, 1–5%, 6–25%, 26–75%, and 76–100%, respectively). The staining intensity was scored as negative (score = 0), weak (score = 1), medium (score = 2), or strong (score = 3). A score ranging from 0 to 12 was calculated by multiplying the staining extent score with the intensity score.

**RNA isolation and quantitative real-time PCR (qRT-PCR)**. Total RNA was extracted with TRIzol (Invitrogen, Carlsbad, CA) following the manufacturer's specifications. Reverse transcription was performed using the Fast Quant RT Kit (Tiangen, China). qRT-PCR was carried out using SYBR Green SuperMix (Roche, Basel, Switzerland) and Roche 480 Real-Time PCR system (Roche, Basel, Switzerland). β-actin or U6 was used as an internal control. The primers of miR-339 and U6 were purchased from GeneCopoeia (GeneCopoeia Inc. Rockville, MD). Primers for miR-339 are 5′-CTAGCTGAGAAGGGGCCACAGGC-3′ (forward) and 5′-ACGCGTCACACTGCATCAGAAGACC-3′ (reverse). Primers for TSPAN15 are 5′-TCCCTCCGTGACAACCTGTA-3′ (forward) and 5′-CCGCCA-CAGCACTTGAACT-3′ (reverse). Primers for β-actin are 5′- TGGCACCCAG-CACAATGAA-3′ (forward) and 5′-CTAAGTCATAGTCCGCCTAGAAGCA-3′ (reverse). Primers for BTRC are 5′-ACCAACATGGGCACATAAACTC-3′ (forward) and 5′-TGGCATCCAGGTATGACAGAAT-3′ (reverse).

**Dual luciferase assay**. The putative seed miR-339-5p recognition sequence from the 3′-UTR of TSPAN15 (GGACAGGG) or with mutations (CCTGTCCC) were separately subcloned into a pEZX-MT06 vector containing the firefly luciferase gene (GeneCopoeia, China) to form two constructs, wt-T15-luc and mut-T15-luc, respectively. The pGL3-TK vector containing the Renilla luciferase gene was used as an internal control. The overexpression constructs, including miR-339-5p and miR-control, were co-transfected with wt-T15-luc or mut-T15-luc into EC109 cells, separately. At 48 h after transfection, the luciferase activity was measured by the Dual-Luciferase Reporter Assay Sysem (Promega Corporation, USA), following the manufacturer's instructions.

**Lentiviral-mediated TSPAN15 overexpression and knockdown**. Lentiviral construct carrying TSPAN15 (GeneCopoeia, China) was packaged in 293FT cells. Virus-containing supernatants were collected and stably transfected into EC109 and EC9706 cells selected by 6 μg mL⁻¹ puromycin (Sigma-Aldrich, St. Louis, MO). Empty vector transfected cells were used as controls. For TSPAN15 knock down assay, two lentiviral constructs containing short hairpin RNAs (shRNA) specifically targeting TSPAN15 were purchased from GeneCopoeia. Either the shTSPAN15 construct or the negative control plasmid was transfected into the 293FT cell line. Virus-containing supernatants were collected for subsequent transduction into KYSE30 cells. Puromycin (Sigma-Aldrich, St. Louis, MO) was used to select for stably transduced cells. The sequences of the two shRNAs against TSPAN15 are as follows: 5′-CCGAGATTGGAGCAAGAAT-3′ and 5′-GCGTTTCAGTGTGCAGGAT-3′.

**Wound healing and invasion assays**. For the wound healing assay, cells were cultured in a 35 mm dish until confluent and then wounded with a sterile tip. The cells were captured at 0, 12, and 24 h after scratching. Invasion assay was performed using BD BioCoat Matrigel Invasion Chambers (Becton Dickinson Labware, Franklin Lakes, NJ) according to the manufacturer's instructions. Cultured cells were trypsinized and resuspended in serum-free DMEM at a density of 60,000 per well onto the upper chambers (8 μm pore size) and DMEM with 10% FBS was added to the lower chambers. After 24 h, cells that had migrated through to the bottom of the insert membrane were fixed, stained with crystal violet and counted under ×20 objective lens. The experiments were repeated thrice.

**In vivo tumorigenesis and metastasis assay**. The study protocol was approved by and performed in accordance with the Committee of the Use of Live Animals in Teaching and Research at Sun Yat-sen University Cancer Center. For in vivo tumorigenic experiment, TSPAN15- and Vec-transfected cells (2×10⁶) were injected into the left and right dorsal flank of 5-week-old athymic nude mice, respectively. Tumor formation in nude mice was monitored over a 4-week period. The tumor volume was calculated by the formula: tumor volume = 0.5 × length × width$^2$. For in vivo metastasis assays, each experimental group consisted of six 5-week-old athymic nude mice. Briefly, 1.5×10⁵ cells were injected

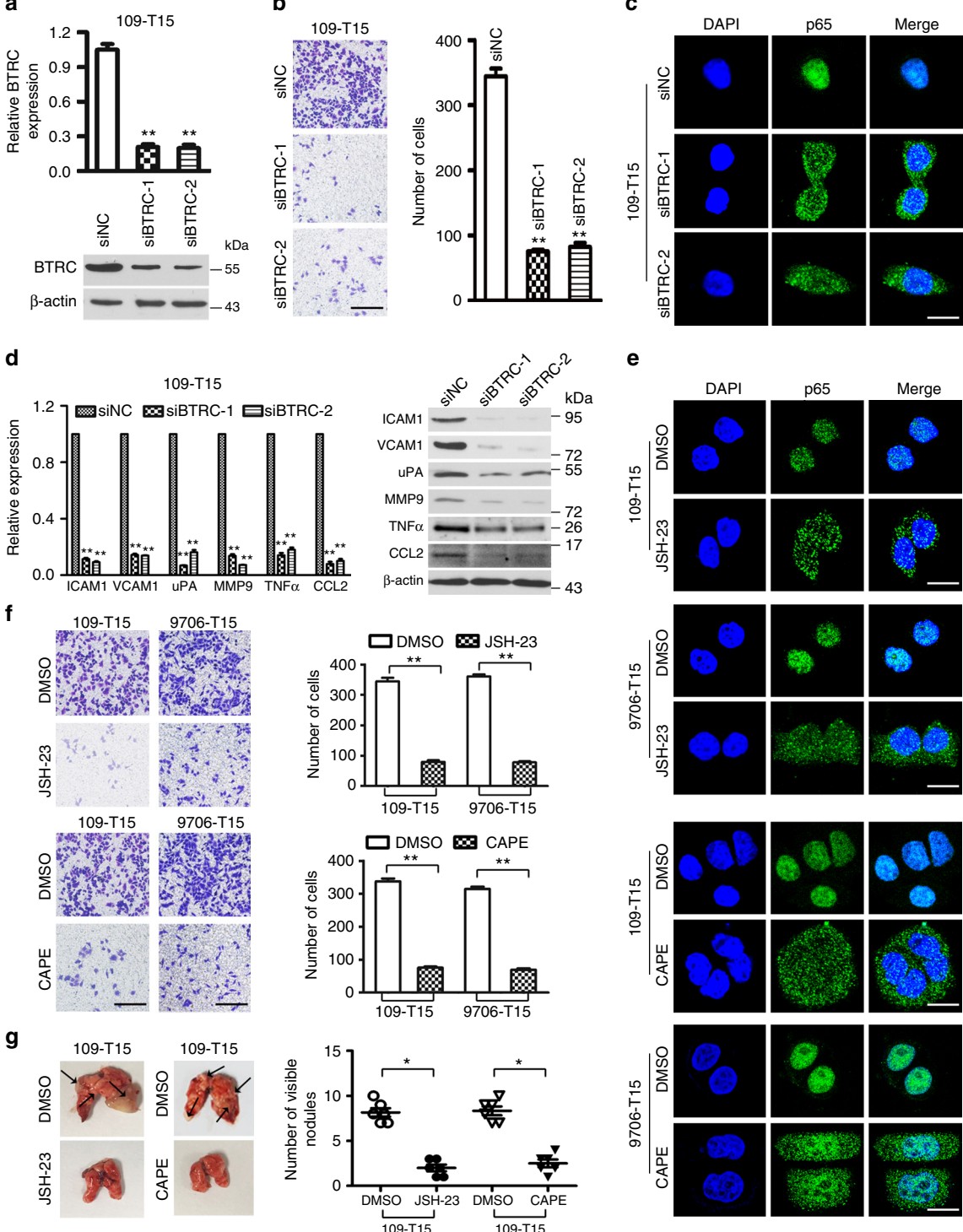

**Fig. 5** Either knockdown of *BTRC* or p65 inhibitors suppresses overexpression of *TSPAN15*-induced cell metastasis. **a** Two siRNAs (si*BTRC*-1 and si*BTRC*-2) against *BTRC* effectively decreased BTRC expression as detected by qRT-PCR and western blotting. Negative control siRNA (siNC) and β-actin were used as negative and endogenous controls, respectively. The data are represented as the mean ± s.d. of three independent experiments. \*\**P* < 0.001, ANOVA with post hoc test. **b** Representatives and summary of invasion assay showing that knockdown of *BTRC* effectively inhibited cell invasion. Scale bar, 200 μm. The values are expressed as the mean ± s.d. of three independent experiments. \*\**P* < 0.001, ANOVA with post hoc test. **c** Knockdown of *BTRC* reduced nuclear location of NF-κB p65 (green). Nuclei were counterstained with DAPI. Scale bar, 20 μm. **d** Knockdown of *BTRC* reduced the mRNA and protein levels of ICAM1, VCAM1, uPA, MMP9, TNFα, and CCL2. The data are represented as the mean ± s.d. of three independent experiments. \*\**P* < 0.001, ANOVA with post hoc test. β-actin served as a loading control. **e** Confocal microscopy of NF-κB p65 staining (green) of JSH-23-treated, CAPE-treated or DMSO-treated *TSPAN15*-expressing cells. Nuclei were counterstained with DAPI. Scale bar, 20 μm. **f** Invasion assay for *TSPAN15*-expressing cells that were treated with JSH-23, CAPE or DMSO. Representative images of invaded cells are shown and the results were summarized respectively. Scale bar, 200 μm. The results are expressed as the mean ± s.d. of three independent experiments. \*\**P* < 0.001, Student's *t*-test. **g** Both JSH-23 and CAPE abrogated TSPAN15-induced elevated cell metastasis in vivo. Black arrow indicates the pulmonary metastatic nodule. The number of metastatic nodules on lung surfaces is summarized. The values were expressed as the mean ± s.d. of six mice. \**P* < 0.01, Student's *t*-test

intravenously through the tail vein into mouse. The mice were killed 8 weeks after injection. Tumor nodules formed on the lung surfaces were macroscopically determined and counted. The lungs were excised and embedded in paraffin. Further, the tissue sections (5 μm) were stained with H&E to visualize the structure.

**Immunoprecipitation (IP) assay.** Whole-cell extracts were prepared in NP-40 lysis buffer (50 mM Tris-HCl pH 7.5, 150 mM NaCl, 5 mM EDTA, 1% NP-40, 0.5% deoxycholate, 0.1% SDS) and incubated with antibodies (anti-TSPAN15 or anti-BTRC) or IgG at 4° C for 2 h, and then mixed with 100 μl protein A/G agarose (Santa Cruz) overnight at 4° C. Beads were washed with PBS three times, and bound protein was denatured with 2× SDS sample buffer. The supernatants were collected and proceeded to SDS-PAGE western blot analysis.

**Mass spectrometry analysis.** Similar to immunoprecipitation assay, anti-TSPAN15 was incubated with cellular protein extracts from KYSE30 cells, and protein A/G agarose beads were then added. Recovered proteins associated with TSPAN15 or IgG were resolved by gel electrophoresis. The bands specific bind to TSPAN15 were excised, and proteomics screening was accomplished by mass spectrometry analysis on a MALDI-TOF-MS instrument (Bruker Daltonics). Full gel of Fig. 3a is provided in Supplementary Fig. 3a.

**Immunofluorescence (IF) staining.** Cells on the coverslips were fixed with 4% paraformaldehyde and incubated with the primary antibody against TSPAN15, FLAG, BTRC or NF-κB p65 at 4 °C overnight. After washing with PBS, cells were then incubated with fluorescence-conjugated secondary antibody (Invitrogen, Carlsbad, CA), and subsequently counterstained with DAPI (Life Technology). Images were captured after staining with anti-fade DAPI solution using a confocal laser-scanning microscope (Olympus FV1000).

**Cycloheximide assay.** Cycloheximide was added into culture medium with the final concentration of 20 μmol L$^{-1}$. Cell lysis were collected at 0, 12, 24, and 36 h after the treatment of cycloheximide.

**Treatment with NF-κB inhibitors.** *TSPAN15*-transfected cells were treated with JSH-23 (200 μM) or CAPE (200 μM) or DMSO for 24 h to examine their effects on nuclear translocation of NF-κB p65, as well as on cell invasion. For in vivo treatment assay, 5-week-old athymic mice were randomly assigned into four groups after injection of 109-T15 cells via the tail vein, each consisting of six mice. They were then treated with JSH-23 (10 μmol kg$^{-1}$ via intragastric administration) or CAPE (10 μmol kg$^{-1}$ via intraperitoneal route) every day. After 8 weeks of treatment, all mice were killed and lung metastases were dissected. DMSO served as a control.

**Western blot assay and antibodies.** Quantified protein lysates were resolved on SDS-PAGE gel, transferred onto PVDF membrane (Millipore, Billerica, MA). Membranes were then blocked with Tris-buffered saline-Tween 20 (TBS-T) containing 5% non-fat milk for 1 h at room temperature and incubated with primary antibody at 4 °C overnight. After washing with TBS-T, membranes were incubated with horseradish peroxidase (HRP)-conjugated secondary antibody. Membranes developed using enhanced chemiluminescence (Amersham Biosciences, Piscataway, NJ). β-actin serves as a loading control. Antibodies used are listed in Supplementary Table 1. Full blots of Fig. 3e are provided in Supplementary Fig. 3b–d.

**Statistical analysis.** No specific statistical tests were used to predetermine the sample size. In vitro and in vivo experiments were monitored in a non-blinded fashion. Statistical analysis was performed using GraphPad Prism 7 software or SPSS version 16.0 (SPSS, Inc., IL). When passing the normality test, a two-tailed Student's $t$-test was used for two-group comparisons, while a two-way ANOVA test with Tukey's post hoc test was used for multiple group comparisons. Paired two-tailed $t$-test was employed to compare the expression of TSPAN15 in primary OSCC tumors and their corresponding adjacent non-tumorous tissues. Kaplan−Meier analysis with log-rank test was applied for survival analysis. The relation between TSPAN15 expression and clinicopathological characteristics was analyzed by Pearson $\chi^2$ test. Univariate and multivariate Cox proportional hazard regression models were used to evaluate the survival hazard using Cox proportional hazard model with a forward stepwise procedure. Differences were considered to be statistically significant when $P < 0.05$ (*$P < 0.01$; **$P < 0.001$).

**Data availability.** The data that support the findings of this study are available from the corresponding author upon request. The proteomic data have been deposited in Figshare.com (https://figshare.com/articles/Proteomic_data-TSPAN15/5868057).

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

## Acknowledgements

This work was supported by grants from the Hong Kong Research Grant Council (RGC) grants including Collaborative Research Funds (C7038-14G and C7027-14G), General Research Funds (HKU/7668/11M, 767313 and CUHK/766613), NSFC/RGC Joint Research Scheme (N_HKU712/12), Hong Kong (CUHK/766613), NSFC/RGC Joint Research Scheme (N_HKU712/12), Hong Kong Health and Medical Research Fund (02133366), the National Basic Research Program of China (2012CB967001), China National Key Sci-Tech Special Project of Infectious Diseases (2013ZX10002-011-005), Young Talent Teachers Plan of Sun Yat-sen University (15ykpy33).

## Author contributions

B.Z.: study concept and design, acquisition of data, analysis and interpretation of data, drafting of the manuscript; Z.Z., L.L., C.J., T.-T.Z., M.-Q.L., D.X. and Y.L.: acquisition of data, technical support; Y.-R.Q., H.L.: material support; Y.-H.Z. and X.-Y.G.: study concept and design, critical revision of the manuscript for important intellectual content and obtained findings.

## Additional information

**Competing interests:** The authors declare no competing interests.

