## [Peer Review File(PDF 275 kb) · Nature Communications]

Reviewer #1 (Remarks to the Author):

Zhang et al presents a comprehensive look at the expression and functional significance of TSPAN15 overexpression in esophageal squamous cell carcinoma.

They show that TSPAN is elevated in esophageal cancer compared to normal tissue and that the extent of elevation correlates with a poorer outcome and metastases. They investigate the mechanism regulating TSPAN levels by providing evidence that TSPAN15 RNA levels are controlled directly by miR-339. Finally they show that TSPAN15 affects cancer cell biology through upregulation of the NFkappaB signalling mediated by binding to BTRC a ubiquitin ligase that targets an inhibitor of NFkappaB. Reduction of TSPAN15 or inhibition of NFkappaB leads to decreased migration and decreased tumor metastasis in an experimental model. Thus they provide a framework in which to consider potential involvement and control of NFkappaB and its regulation in esophageal cancer.

More specifically, they first show in a cohort of 46 patient specimens that TSPAN15 RNA expression is increased in the cancer compared to adjacent esophagus. They confirm this result on the protein level using IHC of clinical specimens. Further comparison of expression levels between cancers showed a correlation of poor outcome and increase lymph node and distant metastasis with higher TSPAN15 staining.

In exploring how TSPAN15 is regulated, the authors first ask how TSPAN15 levels might be regulated and identify miR-339 as a putative control for TSPAN15, then finding that miR-339 expression inversely correlated with TSPAN15 levels. In further experiments they show that manipulating the expression of miR339 alters TSPAN15 levels confirming the suggestion that miR-339 levels play a role in controlling tSPAN15 expression.

The heart of the paper now focuses upon the functional changes to the cancer cells generated by TSPAN15 expression and how they are mediated. They find that TSPAN15 binds BTRC a ubiquitin ligase for I kappa B alpha and that altering TSPAN15 affects NFkappaB signaling, as well as tumor metastasis by intravenous injection, and migration in tissue culture.

The main questions I have are in regard to their evidence that the downregulation of TSPAN15 and NFkappaB signaling reduces expression of some known gene products known to be affected by the NFkappaB signaling pathway such as VCAM, and MMP-9. I would ask whether they also looked at cytokines such as TNF alpha, IL6, CCL2... that are more uniquely NFkappaB dependent and also greatly influence metastatic outcome and tumor inflammation. On that note it appears in Fig. 4 that the staining for ICAM and VCAM and MMP-9 may be in infiltrating leukocytes rather than the tumor cells, which would be consistent with induction of inflammatory cytokines. The authors should note that the VCAM and MMP9 panels in Fig4d appear to be identical.

Minor points.

In the TSPAN15 pull down proteins below 55kDa are shown- why have the higher molecular weights been shown?

The images of cells with immunofluorescence are very small and should be enlarged.

The methods, doses and times for administration of the NFkappaB inhibitors are not indicated.

The tumor growth curve for subcutaneous tumors is a bit limiting as they only show the weights at the end of the experiment. The timing of the endpoint is not clear. One might expect to have volumes measured with time.

Reviewer #2 (Remarks to the Author):

Authors focused on the tetraspanin family in esophageal cancer. These data was significant and informative, however, the reason to picked up the TSPAN15 is unclear. The method of RNAseq and sample information was not described in detail. Moreover, the degradation of I κ B α by BTRC via ubiquitination should be validated in this study. I raised several points to improve the content of the report.

1. In public database, the lists of several tumor specific genes were available. Author should validate the reproducibility of the data in this study because sample size might be too small.

2. Authors described the significance of esophageal cancer and tetraspanin family in introduction section. However, this study includes many oncological factors including microRNA, NF κ B signaling, and BTRC. This introduction is not enough to understand study design. Please explain the several keywords to understand the results for readers.

3. I can not validate the reference 10 to explain the relationship of BTRC and p-IkB. This relationship is most important part in this study because I can not find other paper to clarify the degradation of IkBa via BTRC. Authors should describe the paper in introduction section in detail.

Minor query

1. Are the WB bands black and white mode?

2. In supplementary figure2c, the validation data of BTRC expression and T15 should shown in parallel.

3. The significance of BTRC in this study should described in abstract. Reader can not understand the title from the abstract information.

Title should be summarized to understand the significance of the paper.

A point-by-point response to the Reviewers' comments and suggestions

Reviewer #1 (Remarks to the Author):

Zhang et al presents a comprehensive look at the expression and functional significance of TSPAN15 overexpression in esophageal squamous cell carcinoma.

They show that TSPAN is elevated in esophageal cancer compared to normal tissue and that the extent of elevation correlates with a poorer outcome and metastases. They investigate the mechanism regulating TSPAN levels by providing evidence that TSPAN15 RNA levels are controlled directly by miR-339. Finally they show that TSPAN15 affects cancer cell biology through upregulation of the NFkappaB signalling mediated by binding to BTRC a ubiquitin ligase that targets an inhibitor of NFkappaB. Reduction of TSPAN15 or inhibition of NFkappaB leads to decreased migration and decreased tumor metastasis in an experimental model. Thus they provide a framework in which to consider potential involvement and control of NFkappaB and its regulation in esophageal cancer.

More specifically, they first show in a cohort of 46 patient specimens that TSPAN15 RNA expression is increased in the cancer compared to adjacent esophagus. They confirm this result on the protein level using IHC of clinical specimens. Further comparison of expression levels between cancers showed a correlation of poor outcome and increase lymph node and distant metastasis with higher TSPAN15 staining.

In exploring how TSPAN15 is regulated, the authors first ask how TSPAN15 levels might be regulated and identify miR-339 as a putative control for TSPAN15, then finding that miR-339 expression inversely correlated with TSPAN15 levels. In further experiments they show that manipulating the expression of miR339 alters TSPAN15 levels confirming the suggestion that miR-339 levels play a role in controlling tSPAN15 expression.

The heart of the paper now focuses upon the functional changes to the cancer cells generated by TSPAN15 expression and how they are mediated. They find that TSPAN15 binds BTRC a ubiquitin ligase for IkappaBalph and that altering TSPAN15 affects NFkappaB signaling, as well as tumor metastasis by intravenous injection, and

migration in tissue culture.

The main questions I have are in regard to their evidence that the downregulation of TSPAN15 and NFkappaB signaling reduces expression of some known gene products known to be affected by the NFKappaB signaling pathway such as VCAM, and MMP-9. I would ask whether they also looked at cytokines such as TNFalpha, IL6, CCL2... that are more uniquely NFkappaB dependent and also greatly influence metastatic outcome and tumor inflammation. On that note it appears in Fig. 4 that the staining for ICAM and VCAM and MMP-9 may be in infiltrating leukocytes rather than the tumor cells, which would be consistent with induction of inflammatory cytokines. The authors should note that the VCAM and MMP9 panels in Fig4d appear to be identical.

Our reply:

According to Reviewer's suggestion, we performed qRT-PCR, Western blotting and IHC staining to investigate the expression of TNF α , IL6 and CCL2 in *TSPAN15*-transfected and *TSPAN15*-knockdown cells. The results showed that expressions of TNF α and CCL2 were increased and decreased in *TSPAN15*-expressing cells and knockdown cells, respectively. Additionally, deletion of *BTRC* expression in 109-T15 cells efficiently repressed the expressions of TNF α and CCL2. These data have been incorporated in the "Results" section (the 1st paragraph on page 12 and the 1st paragraph on page 13), Fig. 4b-d and Fig. 5d in the revised manuscript. However, the expression of IL6 was hardly detected in parental EC109, EC9706 and KYSE30 cells, *TSPAN15*-transfected cells or *TSPAN15*-knockdown cells (data not shown).

Thanks for Reviewer's comment. Actually, we found that besides some infiltrating leukocytes, most tumor cells in 109-T15 derived lung nodes indeed exhibited significantly high ICAM1, VCAM1 and MMP9 positive signals. Fig. 4d has been replaced with the images that can clearly display high expression of ICAM1, VCAM1 and MMP9 in tumor cells in the revised manuscript.

Thanks for Reviewer's reminding. The image of VCAM1 in Figure 4d has been replaced in the revised manuscript.

Minor points.

In the TSPAN15 pull down proteins below 55kDa are shown- why have the higher molecular weights been shown?

Our reply:

The gel was split and there were no specific TSPAN15 pull down proteins with molecular weights ≥ 72 kDa, therefore we only presented < 72 kDa proteins. The image of the full gel has been incorporated in Supplementary Fig. 3 in the revised manuscript.

The images of cells with immunofluorescence are very small and should be enlarged.

Our reply:

According to Reviewer's suggestion, the images of IF staining have been replaced by images with higher magnification, which have been incorporated in Fig. 3c, Fig. 4a, Fig. 5c and e in the revised manuscript.

The methods, doses and times for administration of the NFkappaB inhibitors are not indicated.

Our reply:

According to Reviewer's suggestion, the information for administration of the NF- κ B inhibitors has been incorporated in the "Methods" section (the 3rd paragraph on page 23) in the revised manuscript.

The tumor growth curve for subcutaneous tumors is a bit limiting as they only show the weights at the end of the experiment. The timing of the endpoint is not clear. One might expect to have volumes measured with time.

Our reply:

According to Reviewer's suggestion, xenograft tumor growth curves have been incorporated in Supplementary Fig. 1e in the revised manuscript.

Reviewer #2 (Remarks to the Author):

Authors focused on the tetraspanin family in esophageal cancer. These data was significant and informative, however, the reason to picked up the TSPAN15 is unclear. The method of RNAseq and sample information was not described in detail. Moreover, the degradation of IκBa by BTRC via ubiquitination should be validated in this study. I raised several points to improve the content of the report.

1. In public database, the lists of several tumor specific genes were available. Author should validate the reproducibility of the data in this study because sample size might be too small.

Our reply:

Thanks for Reviewer's suggestion. We exploited a web serve GEPIA (Gene Expression Profiling Interactive Analysis) to analyze *TSPAN15* expression in esophageal cancer (ESCA) and normal samples based on TCGA and GTEx data. The result demonstrates that *TSPAN15* expression is obviously higher in ESCA, which is consistent with our RNA-Seq data. The data have been incorporated in the "Introduction" section (the 2nd paragraph on page 4) and Supplementary Fig. 1a in the revised manuscript.

2. Authors described the significance of esophageal cancer and tetraspanin family in introduction section. However, this study includes many oncological factors including microRNA, NFκB signaling, and BTRC. This introduction is not enough to understand study design. Please explain the several keywords to understand the results for readers.

Our reply:

Thanks for Reviewer's suggestion. The information for these keywords, miRNA, NF-κB signaling, and BTRC has been incorporated in the "Introduction" section (the 2nd and 3rd paragraph on page 3) in the revised manuscript.

3. I can not validate the reference 10 to explain the relationship of BTRC and p-IκB. This relationship is most important part in this study because I can not find other

paper to clarify the degradation of IκBα via BTRC. Authors should describe the paper in introduction section in detail.

Our reply:

Thanks for Reviewer's comment. BTRC, also known as β-TrCP and FWD1, has been proved to mediate the degradation of p-IκBα by various studies. This information has been incorporated in the "Introduction" section (the 1st paragraph on page 4) in the revised manuscript. Besides, we validated the effects of BTRC on the degradation of p-IκBα by knockdown *BTRC* in KYSE30 cells. These data have been incorporated in the "Results" section (the 2nd paragraph on page 10) and Supplementary Fig. 2d in the revised manuscript.

Minor query

1. Are the WB bands black and white mode?

Our reply:

Yes.

2. In supplementary figure 2c, the validation data of BTRC expression and T15 should shown in parallel.

Our reply:

According to Reviewer's suggestion, the data of both BTRC and TSPAN15 expressions has been incorporated in Supplementary Fig. 2c in the revised manuscript.

3. The significance of BTRC in this study should described in abstract. Reader can not understand the title from the abstract information.

Title should be summarized to understand the significance of the paper.

Our reply:

Thanks for Reviewer's suggestion. The abstract has been revised according to Reviewer's suggestion in the revised manuscript.

According to Reviewer's suggestion, the title has been changed to "TSPAN15 Interacts with BTRC to Promote Oesophageal Squamous Cell Carcinoma Metastasis via Activating NF- κ B signaling".

Reviewer #1 (Remarks to the Author):

The revisions have addressed my concerns.

Reviewer #2 (Remarks to the Author):

This manuscript is a re-submission for publication. The authors provide additional information and do a job of answering many of the reviewer's critiques of their initial submission. However, this paper is lacking the investigation of fundamental mechanism for TSPAN15 in cancer.

A point-by-point response to the Reviewers' comments and suggestions

Reviewer #1 (Remarks to the Author):

The revisions have addressed my concerns.

--

Reviewer #2 (Remarks to the Author):

This manuscript is a re-submission for publication. The authors provide additional information and do a job of answering many of the reviewer's critiques of their initial submission. However, this paper is lacking the investigation of fundamental mechanism for TSPAN15 in cancer.

Our reply:

Thanks for Reviewer's comment. Our findings demonstrates that TSPAN15 specifically interacts with BTRC to promote the ubiquitination and proteasomal degradation of p-I κ B α , and thereby triggers NF- κ B nuclear translocation and subsequent activation of transcription of several metastasis-related genes, including ICAM1, VCAM1, uPA, MMP9, TNF α and CCL2. In addition, we prove that miR-339-5p directly targets 3'-UTR of *TSPAN15* to suppress *TSPAN15* expression at the transcriptional level. Therefore, we believe that the present manuscript illustrates TSPAN15 oncogenic functions and its fundamental mechanism in OSCC as well.

--